

# Genetic and morphometric divergence in the Garnet-Throated Hummingbird *Lamprolaima rhami* (Aves: Trochilidae)

Luz E. Zamudio-Beltrán[1,2] and Blanca E. Hernández-Baños[2]

[1] Posgrado en Ciencias Biológicas, Universidad Nacional Autónoma de México, Ciudad de México, Mexico
[2] Departamento de Biología Evolutiva, Facultad de Ciencias, Museo de Zoología, Universidad Nacional Autónoma de México, Ciudad de México, Mexico

## ABSTRACT

Cloud forests are one of the most endangered ecosystems in the Americas, as well as one of the richest in biological diversity in the world. The species inhabiting these forests are susceptible to environmental changes and characterized by high levels of geographic structure. The Garnet-Throated Hummingbird, *Lamprolaima rhami,* mainly inhabits cloud forests, but can also be found in other habitats. This species has a highly restricted distribution in Mesoamerica, and five disjunct regions have been delimited within the current geographic distribution of the species from Mexico to Honduras. According to variation in size and color, three subspecies have been described: *L. r. rhami* restricted to the Mexican highlands and Guatemala, *L. r. occidentalis* distributed in Guerrero (Mexico), and *L. r. saturatior*, distributed in the highlands from Honduras and El Salvador. We analyzed the levels of geographic structure in *L. rhami* and its taxonomic implications. We used mitochondrial and nuclear DNA to analyze genetic variation, demographic history, divergence times, reconstructed a multilocus phylogeny, and performed a species delimitation analyses. We also evaluated morphological variation in 208 specimens. We found high levels of genetic differentiation in three groups, and significant variation in morphological traits corresponding with the disjunct geographic populations. *L. rhami* presents population stability with the highest genetic variation explained by differences between populations. Divergence time estimates suggest that *L. rhami* split from its sister group around 10.55 million years ago, and the diversification of the complex was dated ca. 0.207 Mya. The hypotheses tested in the species delimitation analyses validated three independent lineages corresponding to three disjunct populations. This study provides evidence of genetic and/or morphometric differentiation between populations in the *L. rhami* complex where four separate evolutionary lineages are supported: (1) populations from the Sierra Madre Oriental and the highlands of Oaxaca (*rhami*), (2) populations from the highlands of Guerrero (*occidentalis*), (3) populations from the highlands of Chiapas and Guatemala (this is a non-previously proposed potential taxon: *tacanensis*), and (4) populations from the highlands of Honduras and El Salvador (*saturatior*). The main promoters of the geographic structure found in the *L. rhami* complex are likely the Isthmus of Tehuantepec as a geographic barrier, isolation by distance resulting from habitat fragmentation, and climatic conditions during the Pleistocene.

Corresponding author
Blanca E. Hernández-Baños,
behb@ciencias.unam.mx

## INTRODUCTION

Cloud forests are one of the most threatened and biodiverse habitats in the world (*Hamilton, 1995*; *Mulligan, 2010*). In Mesoamerica, the transition zone between the Nearctic and Neotropical regions (*Ríos-Muñoz, 2013*; *Morrone, 2014*), cloud forests are restricted to forest habitats between 600 and 3,000 m above sea level (*Foster, 2001*). Several studies have tried to describe the evolutionary processes that have shaped the enormous diversity observed in cloud forests, concluding that species show high levels of isolation and population differentiation when compared to lowland forest habitats that may be more geographically interconnected (*De Barcellos Falkenberg & Voltolini, 1995*; *Ataroff & Rada, 2000*; *Ornelas et al., 2013*). However, studies at the population level, including a large sampling effort, are crucial to describe intraspecific variation more precisely (*Bonaccorso et al., 2008*; *McCormack et al., 2008*; *Arbeláez-Cortés & Navarro-Sigüenza, 2013*).

Recently, several studies have focused on describing historical patterns and recognizing new species in cloud forests (*González-Rodríguez et al., 2004*; *Cortés-Rodríguez et al., 2008*; *Ornelas, Ruiz-Sánchez & Sosa, 2010*; *González, Ornelas & Gutiérrez-Rodríguez, 2011*). However, the fast pace at which these forests are disappearing due to anthropogenic causes is one of the multiple reasons to promote the study of the evolutionary processes taking place in this particularly diverse ecosystem (*Olander, Scatena & Silver, 1998*; *Martínez-Morales, 2005*).

The Trochilidae family includes several species complexes that have been excellent models for evolutionary studies (*Bleiweiss, 1998a*; *McGuire et al., 2007*; *McGuire et al., 2014*), though to date, only a few of these studies focus on species inhabiting cloud forests (*Bleiweiss, 1998b*; *Chaves et al., 2007*; *Cortés-Rodríguez et al., 2008*; *Chaves & Smith, 2011*). The Garnet-Throated Hummingbird, *Lamprolaima rhami*, (*Lesson, 1838*) has a restricted Mesoamerican distribution; it can inhabit tropical upland forests, pine-oak forests and scrub, but primarily occupies cloud forest habitats, within an altitudinal range between 1,200 and 3,000 m (*Howell & Webb, 1995*; *Schuchmann & Boesman, 2018*). It is considered a relatively sedentary species, though individuals do show some seasonal movement to higher elevations during the breeding season (*Schuchmann & Boesman, 2018*).

*L. rhami* has a discontinuous distribution with five clearly distinguished geographic areas: (1) Sierra Madre Oriental (Puebla to Veracruz) and the northern highlands of Oaxaca (Mexico), (2) the highlands of Guerrero, in the Sierra Madre del Sur (Mexico), (3) the southern highlands of Oaxaca, in the Sierra de Miahuatlan (Mexico), (4) the highlands of Chiapas (Mexico) and Guatemala, and (5) the highlands of Honduras and El Salvador, in Central America. Three subspecies have been recognized based on differences on size and color: *L. r. rhami*, *L. r. occidentalis*, and *L. r. saturatior*. *L. r. rhami* is restricted to the highlands of central and southern Mexico (in the states of Puebla, Veracruz, Oaxaca and Chiapas) and the Guatemala highlands (*Lesson, 1838*; *Peters, 1945*). *L. r. occidentalis*

corresponds to populations found in a restricted patch in Guerrero, southwestern Mexico (*Phillips, 1966*). *L. r. saturatior* is found in the highlands of Honduras and El Salvador (*Griscom, 1932*; *Peters, 1945*). *Schuchmann & Boesman (2018)* considered that *L. rhami* possibly belongs to the genus *Basilinna*, and argued that *occidentalis* and *saturatior* are not suitable for subspecific recognition, proposing them only as races, considering that traits used in their descriptions are either age-dependent or clinal in character (color and size variation).

Considering its restricted distribution and frequent occupation of highly fragmented forests with unique bioclimatic characteristics, *L. rhami* represents an interesting model to assess evolutionary hypotheses about geographic structure and populations dynamics with conservation implications. Hence, the main objectives of this paper are to: (1) evaluate the genetic and morphometric variation of the *Lamprolaima rhami* complex by comparing individuals from the five regions where it is distributed, (2) describe the phylogenetic relationships within *L. rhami* using a multilocus dataset (nuclear and mitochondrial DNA), and (3) propose a hypothesis of its evolutionary history. Based on the characteristics of cloud forest and the site fidelity of this hummingbird species, we expect to find high levels of genetic structure supported by congruence in morphological variation within the *L. rhami* complex. Thus, phylogenetic discontinuities and spatial disjunction are expected rather than phylogenetic continuity and lack of spatial disjunction.

## METHODS

### Taxon sampling and sequencing

We obtained tissues from 54 individuals of *L. rhami* from 14 localities across most of its geographic range (Table S1, Fig. 1, using field collecting permit from Instituto Nacional de Ecología, SEMARNAT: FAUT-0169). We defined five groups *a priori* to evaluate genetic and morphological variation among the five allopatric regions where the species occurs: (1) the Sierra Madre Oriental (SMO), (2) the highlands of Guerrero (GRO), (3) the Sierra of Miahuatlan in Oaxaca (MIA), (4) the highlands of Chiapas and Guatemala (CHIS), and (5) the region in Central America comprising the highlands of Honduras and El Salvador (CA). Tissue samples were obtained for four geographic groups (excluding the CA group) from the following collections: ''Museo de Zoología Alfonso L. Herrera'' (Universidad Nacional Autónoma de México), Museum of Natural Science (Lousiana State University), and Museum of Vertebrate Zoology (University of California, Berkeley).

DNA was extracted using the DNAeasy[TM] kit (Qiagen Inc., Valencia, CA, USA), following the manufacturer's protocols. To evaluate genetic variation in the complex, two mitochondrial markers were obtained from the 54 samples (Control Region, *CR*; and subunits 6 and 8 from ATPase gene, *ATPase 6 & 8*). To evaluate phylogenetic relationships and for species delimitation analyses between groups, two additional mitochondrial markers and four nuclear regions were surveyed in a subsample of 31 individuals (NADH dehydrogenase subunit 2, *ND2*; NADH dehydrogenase subunit 4, *ND4*; the 7th intron of the beta fibrinogen gene, *BFib*, the regions between exons 4 and 5 of the Muscle Skeletal Receptor Tyrosine Kinase gene, *MUSK*; a segment comprising the end of exon 6 and the

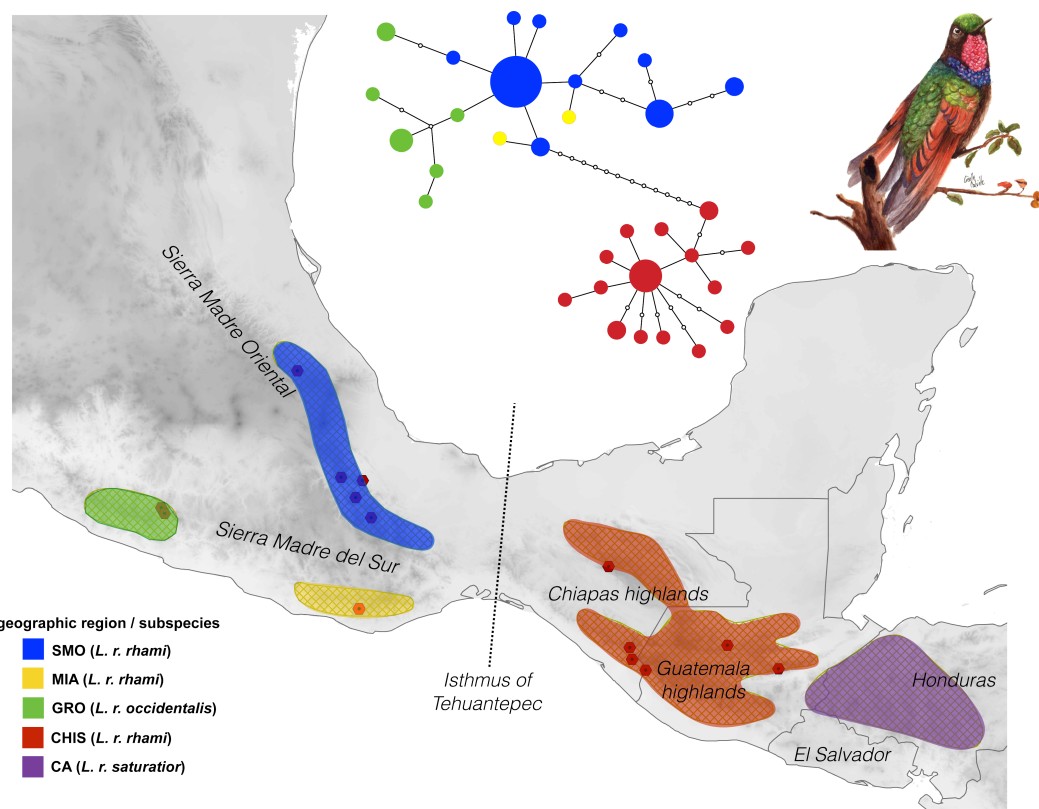

**Figure 1** **Geographic distribution of the *Lamprolaima rhami* complex.** Hexagons represent sampled localities corresponding to tissues used in this study. Geographic groups defined *a priori* are drawn in different colors. Geographic groups: Sierra Madre Oriental (SMO, *blue*), highlands of Guerrero (GRO, *green*), Sierra of Miahuatlan in Oaxaca (MIA, *yellow*), highlands of Chiapas and Guatemala (CHIS, *red*), highlands of Honduras and El Salvador, Cental America (CA, *purple*). The statistical parsimony haplotype network for 54 individuals of *L. rhami* constructed with concatenated mtDNA dataset (ATPase 6 and 8, and control region) is shown above the map. The size of each circle is proportional to the number of individuals carrying each haplotype. Illustration of *L. rhami* by Giselle Fernanda Calvillo García.

beginning of exon 8 of the Ornithine Decarboxylase gene, *ODC,* and intron 5 of adenylate kinase gene, *AK1*). We included sequences from the same molecular markers for *Eugenes fulgens, E. spectabilis*, and *Tilmatura dupontii*, to serve as outgroups (*McGuire et al., 2007*; *Zamudio-Beltrán & Hernández-Baños, 2015*).

We amplified these molecular markers via polymerase chain reaction (PCR) using specific primers and protocols (Table S3). Reactions contained 10× buffer (1.25 μL), 10 mM dNTP (0.19 μL), 50 mM $MgCl_2$ (0.38 μL), 10 μM of each primer (0.25 μL), 0.1 μL of *Taq* (INVITROGEN), and 0.5 μL of genomic DNA (12.5 μL total volume). PCR products were visualized on a 1% agarose gel, and DNA sequencing was performed by the High-Throughput Genomics Unit Service of the University of Washington. We edited and aligned chromatograms with Sequencer v4.8 (GeneCodes Corporation, Ann Arbor, MI, USA). The multilocus alignment can be found in FigShare (https://figshare.com/projects/Lamprolaima_rhami_project_/34487).

## Population structure

To evaluate the number of haplotypes and their relationships, a statistical parsimony haplotype network was constructed for the concatenated dataset of mitochondrial markers (CR and ATPase 6 & 8), using the program TCS v1.21 (*Clement, Posada & Crandall, 2000*).

To analyze genetic diversity and genetic structure, we obtained values of haplotype diversity, nucleotide diversity, mean number of pairwise differences, and population $F_{ST}$ values. These analyses were performed with 1,000 replicates, using the program Arlequin v3.11 (*Excoffier, Laval & Schneider, 2005*). Using the same program, we conducted an analysis of molecular variance (AMOVA; *Excoffier, Smouse & Quattro, 1992*) to detect structure between populations based on comparisons between geographically defined groups. According to our results, regions on both sides of the Isthmus of Tehuantepec were also evaluated (IT, east and west).

To evaluate the isolation by distance among geographic regions, we performed a Mantel Test with 1,000 iterations, comparing matrices of genetic and geographic distances, using the program zt v1.1 (*Bonnet & De Peer, 2002*). Statistical analyses were not performed in the MIA group because of limited number of samples ($n = 2$), but were considered in haplotype networks and phylogenetic analyses.

## Demographic analyses

To evaluate demography and population stability, we obtained Tajima's *D* and Fu's *Fs* values, in Arlequin v2.11 (*Excoffier, Laval & Schneider, 2005*), with 1,000 replicates (mtDNA database). Using the same program, parameters, and database, we further evaluated the historical demography of each group under an expansion model with a MISMATCH distribution test and estimated its significance with the raggedness index (*Slatkin & Hudson, 1991*; *Rogers & Harpending, 1992*; *Harpending, 1994*). To analyze variation in effective population size over time, we used Bayesian skyline plots (BSP; *Drummond et al., 2005*) performed in BEAST v1.6.0 (*Drummond & Rambaut, 2007*), with 10 million steps for mtDNA, using a mean rate of 0.023 substitutions per site per lineage per million years (s/s/l/My), under Control Region and ATPase estimates (*Lerner et al., 2011*).

## Evolutionary models and phylogenetic analyses

For each molecular marker (mtDNA and nuclear DNA), we calculated the evolutionary model that best fit the data based on the Akaike Information Criterion AIC (*Akaike, 1987*) using jModelTest (*Posada, 2008*). We performed a phylogenetic reconstruction using the Bayesian Inference (BI) approach in Mr. Bayes v3.0 (*Huelsenbeck & Ronquist, 2002*). We assigned different evolutionary models to each gene partition. We ran four simultaneous chains for each Monte Carlo Markov Chain analysis for 50 million generations, sampling every 1000 generations. The results were visualized to ensure ESS (effective sample sizes) values higher than 200, and the burn-in value was determined using Tracer v1.6.0 (*Rambaut, Suchard & Drummond, 2013*). The initial 20% of generations were eliminated. The remaining trees were used to construct a majority rule consensus tree with posterior probability distributions, which was visualized using the program FigTree v1.2.3 (http://tree.bio.ed.ac.uk/software/figtree/).

## Morphological variation

To examine morphological variation between groups of *L. rhami*, we took five measurements from 208 voucher specimens (Table S2) corresponding to four of the five geographic groups defined *a priori* (SMO, GRO, CHIS, CA). These specimens were available from the following collections: Museo de Zoología "Alfonso L. Herrera" (MZFC, UNAM), Museum of Comparative Zoology (MCZ), American Museum of Natural History (AMNH), Donald R. Dickey Bird and Mammal Collection (BMC), and the Moore Lab of Zoology (MLZ). Measurements for bill length (from the base to the tip of the upper mandible), bill width (width at the nostrils), bill depth (from the upper mandible to the base of the bill at the nostrils), and wing chord (distance from the carpal joint to the tip of the longest primary) were taken with a dial calliper with a precision of 0.1 mm, while tail length (distance from the uropigial gland to the tip of the longest rectrix) was determined with a milimetric ruler. A single observer took all measurements. Subsets of individuals were measured twice to confirm consistency between measurements using correlation coefficients obtained in STATISTICA v7 (StatSoft, Inc., Tulsa, OK, USA). When variation among measurements was low or null, all voucher specimens were measured once and these values were used in further analyses. To test the normality of our data, we performed a Lilliefors (Kolmogorov–Smirnov) test using the R package Nortest v1.0-4 (*Gross & Ligges, 2015*). This test was conducted with raw and log-transformed data. Since the data were not normally distributed, a Wilcoxon/Mann–Whitney test (*Bauer, 1972*) was performed to evaluate differences between males and females. To evaluate differences among groups, and after confirming differences among sexes, we conducted two sets of Kruskall-Wallis tests (*Hollander & Wolfe, 1973*) to compare: (1) geographic groups (SMO, GRO, CHIS, CA), and (2) groups separated by the Isthmus of Tehuantepec (east, west). All tests were conducted for each variable, treating males and females separately. Statistical analyses were performed using RStudio v.1.1.447 (*RStudio Team, 2016*).

## Species delimitation and divergence times

According to the results, we assess the limits between different groups based on: (1) disjunct populations (groups by geographic region: SMO, GRO, MIA, CHIS), (2) phylogroups (SMO/MIA, GRO, CHIS), and (3) groups separated by the Isthmus of Tehuantepec (east: SMO/MIA/GRO, and west: CHIS). We used the command line of coalescent approach implemented in Bayesian Phylogenetics and Phylogeography software (BP&P v3.4, (*Rannala & Yang, 2003*; *Yang & Rannala, 2010*). This method uses the multispecies coalescent model (MSC) to compare different models of species delimitation (*Yang & Rannala, 2010*; *Rannala & Yang, 2013*) and species phylogeny (*Yang & Rannala, 2014*; *Rannala & Yang, 2017*) in a Bayesian framework, accounting for incomplete lineage sorting due to ancestral polymorphism and gene tree-species tree discordance. We used the concatenated data set of eight molecular markers (mt DNA and nuclear DNA), but mitochondrial markers were treated as one locus, so the total number of this parameter was set to nloci = 5. To confirm consistency between runs, we performed multiple analyses using algorithms 0 and 1 (0: species tree given as fixed, 1: species tree given treated as the guide tree), selecting different seed number between runs and changing finetune parameters

($\varepsilon$, algorithm prior), as suggested by *Yang & Rannala (2010)*. After confirming consistency between runs, the subsequent analyses were performed according to species delimitation using a user-specified guide tree (speciesdelimitation = 1, speciestree = 0), with values of $\varepsilon = 5$. The analyses were conducted using parameter finetune = 1, which allows the program to make automatic adjustments to prior parameters. Because different values of $\theta$ (ancestral population size, the product of effective population size N and mutation rate $\mu$ per site) can result in different posterior probabilities for the same guide tree (*Leaché & Fujita, 2010*; *Yang, 2015*), we used three different values: (1) low $\theta$ priors (0.0001, IG: 3, 0.0002), (2) medium $\theta$ priors (0.001, IG: 3, 0.002), and (3) high $\theta$ priors (0.01, IG: 3, 0.02). The inverse gamma prior to $\tau$ (species divergence times) was set to tauprior = 3, 0.03, 1.5% of sequence divergence. Each analysis was run with the reversible-jump Markov chain Monte Carlo algorithm (rjMCMC) for 100 thousand generations, sampling every five, and discarding 30 thousand generations as burn-in.

To estimate divergence times, we decided to evaluate the hypothesis of three independent groups based on full evidence (phylogroups: SMO/MIA, $n = 12$; GRO, $n = 9$; CHIS, $n = 10$). We used the concatenated data set (mt DNA and nuclear DNA) that included data from *Eugenes fulgens, E. spectabilis*, and *Tilmatura dupontii* as outgroups. For each partition, we assigned the previous selected evolutionary model. This analysis was performed using StarBeast (*Beast; *Heled & Drummond, 2010*). We employed an uncorrelated lognormal relaxed clock and a Yule process speciation model to model the tree prior. We assigned a calibration node based on a secondary calibration obtained for the split between the "Mountain Gems" clade (*L. rhami, E. fulgens*, and *E. spectabilis*) and "Bees" clade (*T. dupontii*; 12.5 Mya; (*McGuire et al., 2014*). We incorporated mean substitution rates reported previously (ATPase 6 and 8, ND2, ND4: *Pacheco et al., 2011*); CR: (*Lerner et al., 2011*); AK1, BFib, MUSK, ODC: (*McGuire et al., 2014*). Three independent analyses were run for 30 million generations, sampling every 1,000. The log and tree files from each analysis were combined in LogCombiner, and visualized in Tracer to confirm convergence and ensure acceptable ESS values (ESS > 200). We discarded the first 15% of trees as burn-in. We used TreeAnnotator v1.8.2 (*Rambaut & Drummond, 2007*) to summarize trees as a maximum clade credibility tree, and to obtain mean divergence times with 95% highest posterior density intervals. The resulting tree was visualized in FigTree v.1.2.3.

## RESULTS

### Genetic diversity and population structure

We obtained a concatenated dataset of 1,402 bp for 54 individuals (527 bp of CR and 875 bp of ATPase 6 & 8). The complementary dataset of five molecular markers for 31 individuals included 875 bp of ATPase 6 and 8, 527 bp of CR, 918 bp of ND2, 521 bp of ND4, 758 bp of BFib, 559 bp of MUSK, 495 bp of ODC, and 416 bp of AK1. The initial dataset included 33 haplotypes (24 found with CR and 15 with ATPase 6 & 8). Estimates of haplotype and nucleotide diversity are presented in Table 1. Overall, high haplotype diversity and low nucleotide diversity were observed within groups (SMO, GRO, CHIS).

The mtDNA network revealed significant population structure within the *L. rhami* complex (Fig. 1). There was a clear separation between populations on either side of

**Table 1** **Statistical parameters of genetic diversity, population structure and population demography for mtDNA (CR, ATPase 6 and 8).** $n$: number of sequences used, $h$: number of haplotypes, Hd: haplotype diversity, $\pi$: nucleotide diversity, Pi: mean number of pairwise differences.

| GROUP | $n$ | $h$ | Hd | $\pi$ | Pi(theta) | Tajima's D | Fu's Fs Test |
|-------|-----|-----|------|--------|-----------|------------|--------------|
| SMO | 22 | 9 | 0.81 | 0.0022 | 4.22 | −0.559 | −5.505[**] |
| GRO | 9 | 6 | 0.89 | 0.0026 | 3.73 | −0.886 | −2.77[*] |
| CHIS | 21 | 15 | 0.94 | 0.0021 | 3.50 | −1.988[*] | −14.93[***] |

**Notes.**
[*]$p$- value < 0.05.
[**]$p$- value < 0.01.
[***]$p$- value < 0.001.

**Table 2** **Population pairwise $F_{ST}$ mtDNA (CR, ATPase 6 and 8).**

|  | SMO | GRO | CHIS |
|------|------|------|------|
| SMO | – | | |
| GRO | 0.176[*] | – | |
| CHIS | 0.769[*] | 0.784[*] | – |

**Notes.**
[*]$p$- value < 0.05.

**Table 3** **AMOVA results on *Lamprolaima rhami* populations defined according to geographic groups, and groups on either side of the Isthmus of Tehuantepec using mtDNA (CR, ATPase 6 and 8).**

|  | $d.f.$ | Sum of squares | Variance components | Percentage of variation | Fixation indices |
|------|--------|----------------|---------------------|-------------------------|------------------|
| **Geographic groups (*a priori*)** | | | | | |
| Among populations | 2 | 167.63 | 5.04 | 76.41 | |
| Within populations | 49 | 76.23 | 1.56 | 23.59 | |
| Total | 51 | 243.87 | 6.59 | | $F_{ST} = 0.76$[***] |
| **Groups on either side of the Isthmus of Tehuantepec** | | | | | |
| Among populations | 1 | 159.05 | 6.28 | 78.74 | |
| Within populations | 50 | 84.82 | 1.70 | 21.26 | |
| Total | 51 | 243.86 | 7.98 | | $F_{ST} = 0.79$[***] |

**Notes.**
[***]$p$- value < 0.0001.

the IT, which were separated by twelve mutational steps, while individuals of GRO were closely linked to those of the SMO. In general, the most frequent haplotype was present in populations from the Sierra Madre Oriental group (SMO). Haplotypes from GRO tended to separate from the main haplotype. In Table 2, $F_{ST}$ values confirm high levels of geographic structure between regions. This further translates into a significant correlation between the genetic distance and the geographic distance matrices, according to the Mantel test, thus suggesting isolation by distance between groups ($r = 0.87$, $p < 0.005$).

AMOVA results indicated that the highest genetic variation was observed among rather than within populations, with similar percentages when grouping populations according to geographic groups or on either side of the IT: 76.41% and 78.74% respectively ($P < 0.0001$, Table 3).

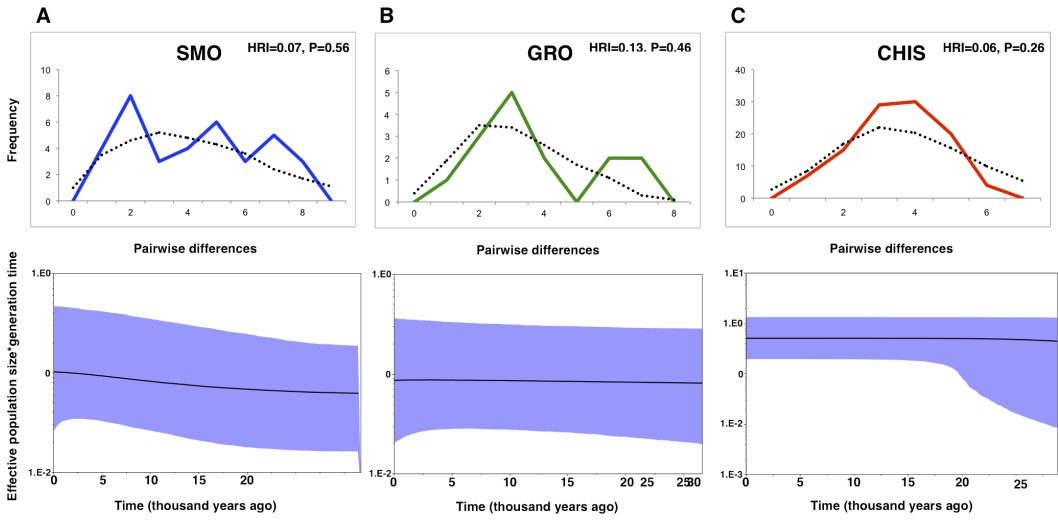

**Figure 2  Mismatch distributions and Bayesian skyline plots for three geographic groups (A) Sierra Madre Oriental, SMO; (B) highlands of Guerrero, GRO; (C) highlands of Chiapas and Guatemala, CHIS) of *L. rhami* (mtDNA: CR, ATPase 6 and 8).** In mismatch distributions, solid lines indicate the observed distributions of pairwise differences, and dotted lines represent simulated distributions under a model of population expansion. In Bayesian skyline plots, solid lines represent median estimates and shaded areas represent 95% confidence intervals.

## Demographic analyses

Differing conclusions among the methods used to evaluate demographic history led to ambiguous results. The occurrence of historical population expansion was supported by negative and significant values of neutrality tests (Tajima's *D* and Fu's *Fs*), except for Tajima's *D* statistic in SMO and GRO groups (Table 1). A mismatch distribution unimodal curve was recovered for CHIS population, but no significant values of the raggedness index indicated possible demographic expansion in all populations, as curves under the expansion model did not deviate from a unimodal distribution. BSP estimates revealed that effective population size was flat across time for GRO. This pattern was also found for CHIS, however, the higher posterior density low interval suggested a growing demographic tendency, and a subtle demographic expansion was recovered in SMO population (Fig. 2).

## Evolutionary models and Phylogenetic analyses

We obtained a concatenated dataset of 5,069 bp. The best-fit models for each molecular marker were as follows: HKY (MUSK), HKY+I (ATPase 6 and 8, ND4), HKY+G (AK1), HKY+I+G (CR), TNR+G (ND2), TPM3uf (ODC), and TPM2uf +I (BFib). Phylogenetic relationships using the multilocus dataset resulted in one main monophyletic group corresponding to individuals from west of the IT (PP > 0.95, Fig. 3: Bayesian Inference). Most individuals from east of the IT were grouped into two well-supported separate clades, but no resolution was recovered for three individuals from this region. Moreover, one well-supported clade included most individuals from GRO group from west of the IT, with the rest of individuals merged in a polytomy with individuals from SMO region.

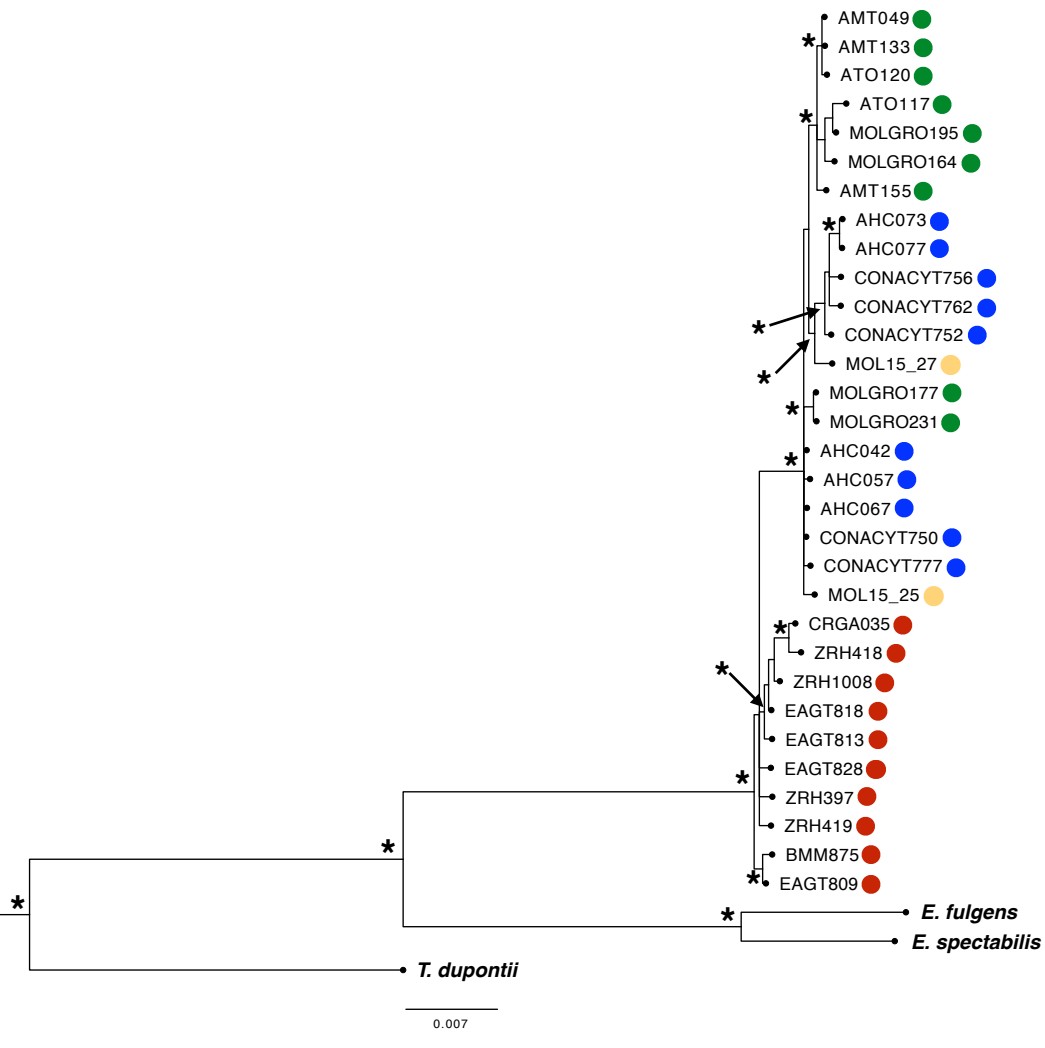

**Figure 3** Phylogenetic Bayesian Inference reconstruction of 31 individuals from the *L. rhami* complex using mitochondrial and nuclear markers (ATPase 6 and 8, CR, ND2, ND4, MUSK, BFib, ODC, and AK1). Posterior probabilities PP > 0.95 are shown (*). Different colors represent different groups according to the geographic regions defined *a priori* (see Fig. 1 legend).

## Morphometric variation

Normality of our data was rejected and there was dimorphism between males and females in all variables (Table S4). General comparisons between geographic groups resulted in significant differences in all variables for both males and females, while comparisons between groups separated by the Isthmus of Tehuantepec showed significant differences in bill width ($\chi^2 = 6.3669$, $p = 0.01163$) and wing chord ($\chi^2 = 8.0642$, $p = 0.004515$) for males; in females all traits were statistically different except for bill length (Table 4). Paired differences between geographic groups (comparing each group against the others), revealed that CA and GRO present significant differences in several traits (Table S4).

**Table 4  Multiple Kruskall–Wallis tests to evaluate differences between groups: (A) by geographic region (SMO, GRO, CHIS, CA), and (B) by groups on either side of the Isthmus of Tehuantepec (IT, east and west).** $p$-value $< 0.05$ (in bold).

|  |  | (A) Geographic region | | | (B) IT | | |
|---|---|---|---|---|---|---|---|
|  |  | $\chi^2$ | df | p-value | $\chi^2$ | df | p-value |
| MALES | Bill length | 7.8661 | 3 | **0.04886** | 1.6847 | 1 | 0.1943 |
|  | Bill width | 19.386 | 3 | **0.0002275** | 6.3669 | 1 | **0.01163** |
|  | Bill depth | 26.425 | 3 | **7.77E–06** | 0.91719 | 1 | 0.3382 |
|  | Wing chord | 19.14 | 3 | **0.0002558** | 8.0642 | 1 | **0.004515** |
|  | Tail length | 16.982 | 3 | **0.0007127** | 1.834 | 1 | 0.1757 |
| FEMALES | Bill length | 9.4421 | 3 | **0.02396** | 0.40845 | 1 | 0.5228 |
|  | Bill width | 13.125 | 3 | **0.004374** | 6.346 | 1 | **0.01176** |
|  | Bill depth | 20.159 | 3 | **0.0001574** | 5.3064 | 1 | **0.02125** |
|  | Wing chord | 29.698 | 3 | **1.60E–06** | 18.547 | 1 | **1.66E–05** |
|  | Tail length | 28.092 | 3 | **3.48E–06** | 10.168 | 1 | **0.001429** |

## Species delimitation and divergence times

Three different species hypotheses were assessed: (A) species delimited by disjunct populations (groups by geographic region: SMO, GRO, MIA and CHIS), B) phylogroups (SMO/MIA, GRO and CHIS), and (C) groups separated by the Isthmus of Tehuantepec (east: SMO/MIA/GRO, and west: CHIS), using different values of $\theta$ priors (population size parameters) to test the sensitivity of the species delimitation results. BP&P analyses testing the first hypothesis (that allopatric populations are different species) resulted in low statistical support for the split of groups SMO and MIA, and suggested the split of GRO group as a valid species only when using low $\theta$ priors. The high probabilities for an ancestral node suggest splitting into multiple species, as well as the validity of CHIS as an independent group (Fig. 4A). Analysis of the three lineages corresponding to phylogroups resulted in high speciation probabilities in all cases, except when using high $\theta$ priors (0.01, IG: 3, 0.02) for the splitting of GRO and SMO/MIA groups (Fig. 4B). Analysis of the two-lineage hypothesis (i.e., two groups separated by the Isthmus of Tehuantepec) suggested that these groups are separate species (Fig. 4C). Varying the $\theta$ priors affected the resulting speciation probabilities; higher $\theta$ priors resulted in lower probabilities than lower $\theta$ priors. Our divergence time estimates (Fig. 4D) showed that the split between *L. rhami* complex and its sister group (genus *Eugenes*) was around 10.55 Mya (8.28–13.14 Mya). The estimate for the first split within the *L. rhami* complex was dated ca. 0.207 Mya (0.091–0.317 Mya), corresponding to the divergence between populations on either side of the IT. The split between the groups from west of the IT (SMO/MIA and GRO) was dated at ~0.087 Mya (0.035–0.146 Mya). The independence of groups was supported by high values of posterior probability.

## DISCUSSION

Our study provides evidence of high levels of genetic and morphometric differentiation among most of the disjunct populations comprising the *L. rhami* complex. Our results

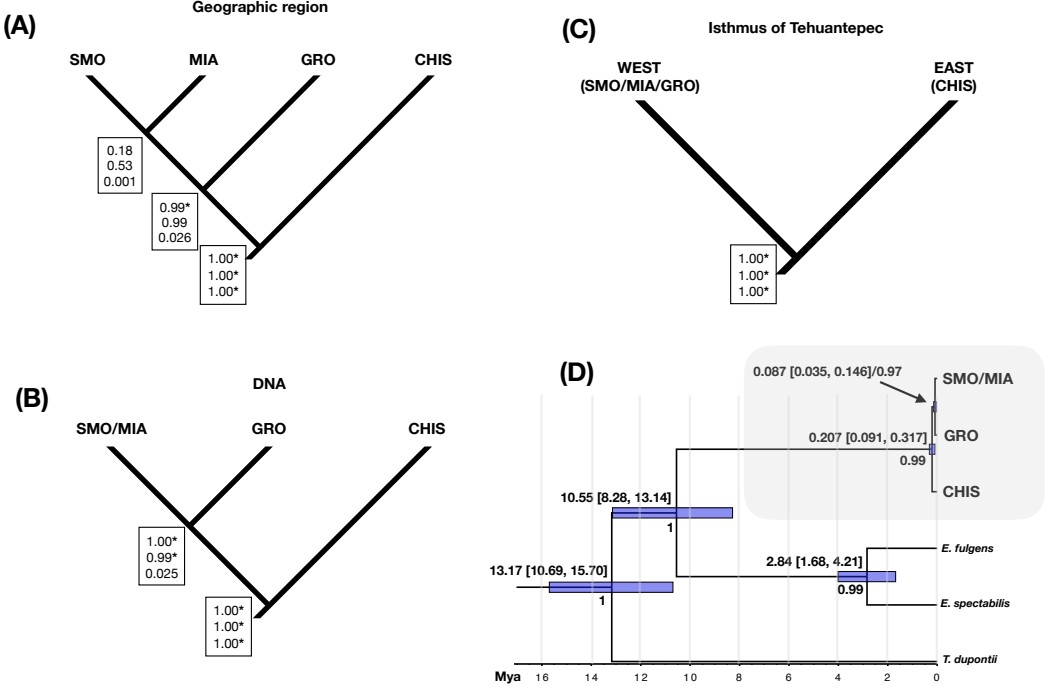

**Figure 4 Species delimitation and divergence times.** Results from a coalescent-based species delimitation analysis implemented in Bayesian Phylogenetics and Phylogeography (BP&P) of three possible hypotheses based on: (A) disjunct populations (groups by geographic region: SMO, GRO, MIA, CHIS), (B) DNA groups (SMO/MIA, GRO, CHIS), and (C) groups separated by the Isthmus of Tehuantepec (east: SMO/MIA/GRO, and west: CHIS). Speciation probabilities for each node are shown in boxes: top, low $\theta$ prior (0.0001); middle, medium q prior (0.001); bottom, high q prior (0.01). * Indicates statistical support for splitting (posterior probabilities of 0.95 or higher). (D) Divergence times and Bayesian species tree topology (*BEAST) for *L. rhami* complex, bars on each node represent 95% of high posterior densities of divergence times (HPD), Mya (Million years ago). Posterior probabilities are shown below node ages.

support the existence of four separate evolutionary lineages: SMO/MIA (*rhami*), GRO (*occidentalis*), CHIS (new suggested taxon: *tacanensis*), and CA (*saturatior*). The AMOVA and $F_{ST}$ values also indicate the presence of strong population structure between these same geographic regions (e.g., 76.41% variation among populations). We found significant morphological differences between the southern populations (Honduras and El Salvador), which we expect to be reflected in genetic differentiation, though we were unable to test this hypothesis here due to a lack of tissue samples.

Morphometric variation in the 208 specimens of *L. rhami* also showed geographic structure among the compared areas (four of the five regions were defined *a priori*). Although the group sampled in the highlands of Chiapas and Guatemala (CHIS) was the most genetically differentiated, the populations of Guerrero (GRO) and Central America (CA) were the most different in morphometric traits. We had no access to genetic samples from the CA region (Honduras and El Salvador), so we cannot confirm if this morphological variation is consistent at the genetic level. Also, we did not have access to enough voucher specimens from MIA to conduct a reasonable morphological statistical analysis. A larger

sampling effort in the southern highlands of Oaxaca (Miahuatlan, Mexico) and in Central America (Honduras and El Salvador) will allow evaluation of species limits in these regions.

The genetic variation found in the *L. rhami* complex corresponds to a phylogenetic discontinuity and a spatial vicariance pattern (*Avise et al., 1987*) resulting from long-term isolation and/or restricted gene flow among groups, probably promoted by geographic barriers. This pattern of high genetic differentiation is congruent with other studies of Mesoamerican vertebrate species (*Barber, 1999*; *Zarza, Reynoso & Emerson, 2008*; *Bonaccorso, 2009*; *Bryson et al., 2011*; *Smith et al., 2011*; *Arbeláez-Cortés, Roldán-Piña & Navarro-Sigüenza, 2014*; *Castañeda Rico et al., 2014*). In Trochilidae, high levels of geographic structure have been previously reported, related to differences in current or historical ecological conditions (*Adelomyia melanogenys*: *Chaves et al., 2007*; *Lampornis amethystinus*: *Cortés-Rodríguez et al., 2008*; *Ornelas et al., 2016*). Also, moderate levels of differentiation have been found in hummingbird species co-distributed in Mesoamerican cloud forests (*Campylopterus curvipennis*: *González, Ornelas & Gutiérrez-Rodríguez, 2011*; *Amazilia cyanocephala*: *Rodríguez-Gómez, Gutiérrez-Rodríguez & Ornelas, 2013*). As expected, the levels of genetic variation were correlated with a pattern of isolation by distance associated with the patchy distribution of cloud forests, where particular environmental characteristics have been reported as drivers of differentiation between populations (*Ramírez-Barahona & Eguiarte, 2014*). In the case of populations from west of the Isthmus of Tehuantepec, geographic structure could be explained by limited gene flow between regions (SMO, MIA and GRO) promoted by isolation by distance. In contrast, the genetic separation between populations on either side of the Isthmus of Tehuantepec is almost certainly influenced by this geographic barrier in addition to distance.

Many phylogeography studies have shown the influence of the Isthmus of Tehuantepec as a driver of isolation in Mesoamerican species. This valley in southeastern Mexico is located near three tectonic plates—North American, Cocos and Caribbean—resulting from different tectonic episodes that took place in the Late Miocene (*Barrier et al., 1998*). Two main diversification events across the Isthmus of Tehuantepec were detected in the regional bird fauna, placing both events within the Pleistocene (*Barber & Klicka, 2010*). It is clear that ecological conditions in this area act as a geographic barrier limiting gene flow of *L. rhami* populations across the Isthmus, and the lack of shared haplotypes across this area demostrates that the Isthmus of Tehuantepec is better considered a hard rather than soft barrier as has been proposed in co-distributed species (e.g., *Amazilia cyanocephala*; *Rodríguez-Gómez, Gutiérrez-Rodríguez & Ornelas, 2013*).

Divergence time estimates show that the split between putative lineages was very recent. The lack of reciprocal monophyly in the multilocus phylogenetic reconstruction (Bayesian inference) may be influenced by low nuclear marker signal due to the temporal scale, resulting in a failure to reconstruct discrete clades in recently evolved species (Knowles & Carstens, 2007). This signal of ancestral polymorphism and/or incomplete lineage sorting is taken into account in BP&P estimates, where information of the chosen molecular markers is fully used in closely related species (*Yang, 2015*). Results of species delimitation were sensitive to different $\theta$ priors, favoring lumping with higher values, as has been previously reported (*McKay et al., 2013*). However, splitting of phylogroups was

consistently supported in most cases. These findings reveal the potential for recognizing at least three distinct cryptic species, also supported by other criteria evaluated here (morphometric differences, population structure and geographic isolation).

Our evaluations of demographic history used different methods (neutrality tests, mismatch distributions and BSP), showing ambiguous patterns of populations dynamics. Range expansion was revealed in the basal group CHIS (Tajima's $D$ and Fu's $Fs$, mismatch), and subtle population size changes over time were detected by the BSP approach. Additionally, expansion signal was not fully supported in the SMO group, and population stability was found in the youngest clade, GRO. Our divergence time estimates provide evidence of recent Pleistocene diversification of the *L. rhami* complex, with the first population split occurring in the Isthmus of Tehuantepec (0.207 Mya, 0.091–0.317 Mya), followed by subsequent separation of groups west of the Isthmus, resulting in the splitting of the SMO/MIA and GRO groups (0.087 Mya, 0.07–0.21 Mya). These processes took place during the Pleistocene, when climatic fluctuations resulted in the expansion and contraction of the ranges of highland species, promoting allopatric differentiation (*Still, Foster & Schneider, 1999*). Our estimates of recent splitting support the hypothesis of differentiation promoted by climatic oscillations rather than by older events related to the complex volcanic history of Mexican highlands, such as mountain uplift.

Two demographic scenarios for cloud forests species during Pleistocene have been proposed, and they corresponded to the dry refugia and the moist forests hypotheses (*Ramírez-Barahona & Eguiarte, 2013*). The dry hypothesis (*Haffer, 1969*) suggests that during glacial cycles, climatic oscillations in cloud forests displaced them downslope into refugia, forcing populations to contract their ranges, which subsequently expanded and recolonized them during interglacial cycles. The moist forest hypothesis suggests that unchanging precipitation conditions did not reduce cloud forests into refugia but favored downslope altitudinal migration, where adapted species expanded their ranges during glacial cycles (connectivity), and fragmented them into higher altitudes during interglacial periods. Historical signals of demographic expansion and high levels of structure are more related to dry refugia, while low levels of structure are consistent with moist forest model due to processes of recurrent population connectivity. Therefore, the high levels of genetic structure found in *L. rhami* are consistent with the dry refugia model. Population dynamics in CHIS were led by down-slope fragmented ranges (glacial cycles) and expanding up-slope (interglacials) revealed by the unimodal distribution of allele differences. In contrast, the SMO and GRO populations show no clear evidence of expansion, and seem to have been maintained *in situ*, a hyphothesis first supported by the shrub *Moussonia deppeana*, a Mesoamerican cloud-forest adapted species (*Ornelas & González, 2014*).

Despite the well-known movement abilities of Trochilidae species, some studies have found that geographical barriers are crucial in promoting high levels of differentiation and the diversification of independent evolutionary lineages in various regions, such as the Andes region (e.g., *Adelomyia melanogenys*, *Chaves & Smith, 2011*), Mesoamerica (*Ornelas et al., 2016*), the Motagua fault region (*Rodríguez-Gómez & Ornelas, 2014*), and the Isthmus of Tehuantepec (*Cortés-Rodríguez et al., 2008*; *González, Ornelas & Gutiérrez-Rodríguez, 2011*). In contrast, in lowland Neotropical birds, high levels of intraspecific diversification

are better explained by the hypothesis of limited dispersal ability (*Burney & Brumfield, 2009*). *L. rhami* exhibits some altitudinal movements related to the presence of resources available along elevation gradients (*Schuchmann & Boesman, 2018*), but long-distance dispersal has not been reported for this species, so both geographic barriers and limited longitudinal and latitudinal dispersal movements could be influencing the geographic separation we observed.

Earlier taxonomic studies described different subspecies for this complex: *L. r. rhami* (*Lesson, 1838*; *Peters, 1945*), *L. r. occidentalis* (*Phillips, 1966*), and *L. r. saturatior* (*Griscom, 1932*; *Peters, 1945*). Our study supports the taxonomic validity of the *occidentalis* (based on genetic and morphometric data) and *saturatior* (based on morphometric data) groups. Thus, the suggestion of considering *occidentalis* and *saturatior* taxa as races (*Schuchmann & Boesman, 2018*) should be reevaluated.

The original description of *L. r. rhami* comprises populations in the highlands of central and southern Mexico, including populations in the state of Guerrero, and populations in Chiapas and Guatemala, which, in agreement with our study, belong to different lineages. Therefore, *rhami* will just include populations from the Sierra Madre Oriental and the highlands of Oaxaca, while *occidentalis* belongs to populations distributed in the highlands of Guerrero. Finally, populations in Chiapas and Guatemala belong to a new suggested taxon (*tacanensis*). The Honduras and El Salvador populations belong to *saturatior,* for which the differentiation is supported by our morphometric results. Supported by our multilocus phylogenetic approach, by the species delimitation estimates, and by the differences in morphometric traits, we found that *L. rhami* is a complex formed by four groups that correspond to separate evolutionary lineages and that should be treated as full species. The importance of delimitation of taxonomic units increases given the level of threat that is reported for the cloud forests of Mesoamerica (resulting from the growth of agricultural and urban areas), and the restricted geographic distribution of *L. rhami*. The problem of an incorrect placement of subspecies is that this could promote underestimation of biodiversity, and therefore mismanagement in conservation efforts (*Zink, 2004*).

## CONCLUSIONS

This study presents clear evidence of morphometric and genetic differentiation between populations of the hummingbird *Lamprolaima rhami*. Pleistocene historical events, the influence of the Isthmus of Tehuantepec as a geographical barrier, and the effects of isolation by distance have shaped the geographical structure found in the *L. rhami* complex. Contemporary habitat fragmentation and the unique bioclimatic characteristics of cloud forests are probably still influencing this pattern of isolation between populations. In general, we found that species diversity within the *L. rhami* complex is currently underestimated, and four taxa should be recognized: (1) populations from the Sierra Madre Oriental and highlands in Oaxaca (*rhami*), (2) populations from highlands in Guerrero (*occidentalis*), (3) populations from highlands in Chiapas and Guatemala (new suggested taxon: *tacanensis*), and (4) populations from highlands in Honduras and El Salvador

(*saturatior*). This study emphasizes the importance of evaluating multiple characters in species complexes that presumably diverged recently or are in an incipient process of speciation.

## ACKNOWLEDGEMENTS

We thank the Museo de Zoología Alfonso L. Herrera (MZFC, UNAM), Museum of Natural Science (LSU), Museum of Vertebrate Zoology at University of California Berkeley (MVZ), D Dittman (LSU), C. Cicero (MVZ), and R. Bowie (MVZ) for provided tissues samples and logistical help; the Moore Lab of Zoology at Occidental College (MLZ), the Bird and Mammal Collection at the University of California Los Angeles (UCLA), the American Museum of Natural History (AMNH), the Museum of Comparative Zoology at Harvard University (MCZ), J. McCormack (MLZ), W. Tsai (MLZ), J. Maley (MLZ), K. Molina (UCLA), P. Sweet (AMNH), L. Garetano (AMNH), J. Trimble (MCZ), K. Eldridge (MCZ), for provided assistance and help in carrying out morphological measurements on museum specimens; A. Gordillo, Isabel Vargas, S. Robles, and F. Rebón for technical help; G.F. Calvillo for drawing *L. rhami* (Fig. 1), and to all the collectors at MZFC. J. J. Morrone, and C. Cordero provided helpful comments that improved the manuscript. L. Kiere reviewed the English.

### Funding

J. Cracraft and the Department of Ornithology at AMNH provided a Collection Study Grant to Luz E. Zamudio-Beltrán. This research was supported by the Posgrado en Ciencias Biológicas (PCBIOL, UNAM). Luz E. Zamudio-Beltrán was supported with the scholarship number 262114/220280 provided by Consejo Nacional de Ciencia y Tecnología (CONACyT, Mexico). This paper is part of the doctoral thesis of Luz E. Zamudio-Beltrán. This paper was written during the sabbatical leave of Blanca E. Hernández-Baños at the Museo Nacional de Ciencias Naturales, Madrid with the support of PASPA/DGAPA UNAM and CONACyT (Registro 613185, Solicitud 45914). The funders had no role in study design, data collection and analysis, decision to publish, or preparation of the manuscript.

### Grant Disclosures

The following grant information was disclosed by the authors:
J. Cracraft and the Department of Ornithology at AMNH.
Posgrado en Ciencias Biológicas (PCBIOL, UNAM).
Consejo Nacional de Ciencia y Tecnología (CONACyT, Mexico): 262114/220280.
PASPA/DGAPA UNAM.
CONACyT: Registro 613185, Solicitud 45914.

### Competing Interests

The authors declare there are no competing interests.

## Author Contributions

- Luz E. Zamudio-Beltrán conceived and designed the experiments, performed the experiments, analyzed the data, prepared figures and/or tables, authored or reviewed drafts of the paper, approved the final draft.
- Blanca E. Hernández-Baños conceived and designed the experiments, performed the experiments, contributed reagents/materials/analysis tools, authored or reviewed drafts of the paper, approved the final draft.

## Field Study Permissions

The following information was supplied relating to field study approvals (i.e., approving body and any reference numbers):

Field experiments were approved by Secretaria del Medio Ambiente y Recursos Naturales (Mexico), scientific collecting permit FAUT-0169.

## Data Availability

Figshare: https://figshare.com/projects/Lamprolaima_rhami_project_/34487.

## Supplemental Information

Supplemental information for this article can be found online at http://dx.doi.org/10.7717/peerj.5733#supplemental-information.

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
