# Peer review of "Genetic and morphometric divergence in the Garnet-Throated Hummingbird Lamprolaima rhami (Aves: Trochilidae)"

_PeerJ, doi:10.7717/peerj.5733_

## Round 0.1 · original submission · Major Revisions

All three reviewers suggested that your paper needs significant improvements before publication. They listed several points with your analyses and data presentation that you should address in the new version. All reviewers pointed out that language and style need to be improved. I strongly recommend that you get editing help from someone with full professional proficiency in English.

When resubmitting your manuscript, please carefully consider ALL points mentioned in the reviewers' comments, explain every change made, and provide proper rebuttals for any remarks not addressed.

·

Basic reporting

Comments to the Authors
The authors, present a robust data set, including morphometric, Phylogeography and multilocus phylogeny, to describe patterns of dispersal, origin and taxonomic affinities. I applaud the authors for the vast geographic examination of specimens to provide a complete picture of the evolution and phylogeography of this group. I think the information provided in this manuscript will serve to provide further evidence to the role of the diversification of Mesoamerica biota.
I, nevertheless, found a series of weaknesses in the manuscript, which should be addressed upon before acceptance. The main problems of the manuscript are related to the morphometric analyzes. The absence of a map, which provides to the reader a notion of spatial distribution of morphometric samples and some methodological steps of morphometric analyzes that were not adequately addressed (detailed below). In addition, non-described details of the phylogenetic analysis and problems in the presentation of some figures, compromise the evaluation of the results and the understanding of the manuscript.
I think, addressing these problems, could set the paper in a more robust way.
Besides these major concerns, I have several comments:

Experimental design

Introduction:

Lines 70 – 73: The current phrasing makes comprehension difficult. Moreover, I don't believe that anthropogenic impacts have an influence on increasing the number of studies, as was suggested.

Line 93: The taxonomic review should be improved. If L. r occidentalis was described as a race, how does it have a subspecies name? This needs to be better defined.

Methods:

Line 124: Names of geographic regions should be carefully reviewed to avoid confusion. (i. e. on line 124 is “SMO&nSMS” and in the figures is “SMO&SMSn”).

Lines 196 – 200: The general ornithological literature has used independent runs, with larger numbers of generations and implemented alternative methods of convergence check (for instance ESS values). I recommend a full review of these settings.

Lines 204 – 207: As above, only one run and no convergence check method was implemented!

Lines 212 – 213: Why so many species within outgroup? This is entirely beyond the scope of the manuscript, and you make don't use these outgroup information in the discussion. I suggest to restrict the outgroup species to the genera Eugenes and Heliomaster.

Line 215: On a phylogenetic scale, such as the one you are using, the Beast manual indicates the Yule-process speciation model. Please explain why you used Birth-Death process speciation model.

Line 228: I think you mean: “four of the five”

Lines 242 – 252: I did not understand why to use two different statistical methods to test the differences between groups. ANOVA and DFA do the same thing differently, so why use both methods?
DFA and ANOVA are based on the normality of the data, you did not present any normality test (i.e. Kolmogorov test)
Zoological literature has extensively used data transformations in order to obtain a better fit. I indicate that you make exploratory tests with these transformations (i.e. log-transformation or square-root transformation).

Line 251: Provide the correct citation to the R software (See indication on the own program)

Validity of the findings

Results:

Line 290 and 292: Cite the acronyms of the geographical locations consistently; omissions of letters may cause confusion in the reader.

The scales used in the trees of Figures 3 and 4 make it impossible to visualize the details of the branches of the tree and compromise the interpretation of the results. It is necessary modify the presentation of these figures!

Line 313: In fact 0.61 Mya, refers to the date of the first split in L. rhami Complex, not to the estimated date for the origin of the clade.

Discussion:
Line 343: I suggest you replace the word phenotypic by morphometric, since you only evaluate a very specific aspect of the phenotype. In the paper, very important aspects of the phenotype, such as plumage and voice, are not considered. This suggestion also applies to the title of the paper.

Line 385 - 386: Which result supports this idea? You should expand the explanation.

Line 397 – 400: This paragraph is speculative; you have no evidence of this in your results. The speculative character of the statement must be explicit. It would be interesting to include some literature that supports this idea.

Line 423: You should change the sentence: “ Provides the existence of occidentalis”. I suggest: “ Supports the taxonomic validity of occidentalis”

Conclusion:

Your conclusions should be expanded and improved. I suggest that you add the taxonomic findings at the conclusion.

Additional comments

The keywords are almost identical to those of the title; you could improve the index by replacing some of those keywords.

I suggest the use of a species delimitation program (i.e. BPP)

Legend Table 1: There is no P in this table!

Trees 3 and 4 need to be redone!

The color schemes, in the morphological and molecular analyzes, could be identical to facilitate the identification by the reader!

Dear Authors, I did enjoy reading your manuscript and thank you guys for your effort to bring novel information on this interesting Hummingbird. Please, read my comments considering that I have a constructive intention to your ideas.
Regards,

·

Basic reporting

The basic structure of the submission meets basic standards. However, the English - though professional - has grammatical and style insufficiencies. As I note in my comments to the authors below, I appreciate that the authors do not speak English as a first language. I would be willing to help if asked to review again, but only if given access to a Word document.

Experimental design

Most of the experimental design is appropriate but I have suggestions for additional analyses that I believe would improve the study and I note sections where insufficient methodological detail was provided.

Validity of the findings

I'm confused by the two phylogenetic estimates that disagree with one another, but the general findings are valid. I make suggestions for improvement in this regard in my General Comments below.

Additional comments

In this manuscript, the authors collect multilocus genetic data and morphometric data to assess levels of genetic and phenotypic divergence in the hummingbird species, Lamprolaima rhami. The genus and species are monotypic, though races (occidentalis and saturatior) were previously designated. Schuchmann (in Handbook of Birds of the World) argued that these races are based on clinal or age-related features and were not suitable for subspecific recognition. Although it would be nice to see these arguments addressed in the Intro, I certainly see the value of a species delimitation study that tests these prior hypotheses as well as the possibility of genetic and phenotypic structure matching the allopatric populations (that evidently don't align perfectly with the ranges of the races).

The data collected for this study are appropriate and sufficient for a paper in PeerJ, however, this manuscript is not quite ready for prime time (in other words, publication). In addition to the technical issues that I will discuss below, the MS requires a fair amount of stylistic and grammatical improvement. I sympathize with the authors, who are not writing in their mother tongue, but there were more stylistic and grammatical changes needed than I could handle while working off of a pdf doc. With a Word document version, I would be willing to roll up my sleeves and help, but it is simply unwieldy to make such corrections on a pdf.

In terms of major issues, I think the largest issue is that authors should consider adding a BPP analysis to the study. Given their interest in species delimitation, they should employ a method that will allow them to assess lineage status in a coalescent framework. I found the description of the *BEAST analysis to be inadequate. The authors did not prove relevant details about the analysis that they did undertake. For example, did they run just one analysis with 3 a priori designated species? If so, why only three? Shouldn’t all combinations of possible numbers of species been under consideration? BPP allows for a simultaneous assessment of species/lineage status as well as relationships. They also provided no basic information about the assumptions of the *BEAST analysis such as models of nucleotide substitution, how many individuals were considered per a priori designated species, etc.

Another issue in the MS is the presentation of two alternative phylogenetic trees. In Figure 3, none of the putative lineages are monophyletic. In the Figure 4 time-tree, two of the three groups are indicated as monophyletic and this tree is then used as the basis for lineage status discussion as well as age estimation for divergences. Why do these two Bayesian trees, presumably based on the same data, differ in such important ways? Minimally, this discrepancy must be addressed before I could envision this manuscript being ready for publication. Also, it seems like a *BEAST time tree is the way to go once the number of lineages is settled upon.

The morphology/phenotype analysis also is poorly elaborated upon in both the Results and Discussion. In addition to the basic problem that there is no key to the letters used to reflect biogeographical regions, it is not clear how we should interpret overlap or lack thereof on the graphs. I was confused by these analyses and I’m sure other readers would be as well.

The Discussion indicates that there is evidence for three genetic lineages. I don’t see it. Depends on whether the tree in figure 3 or figure 4 is assumed. The morph results are not described well enough to interpret lineage status on the basis of phenotype.

Finally, the authors did not come to a decision on the correct number of species for this group. I also thought that they didn’t really spend much time discussing the implications of their findings for cloud forest habitats and their role in speciation. Granted, they mentioned cloud forests in the Discussion, but not to the extent that I think could be warranted depending on how the genetic data sort themselves out. I find that there remains uncertainty about the lineage status of these hummingbird populations and that resolving that uncertainty will allow this study to have a larger impact.


Custom checks:

1) I did not see a data deposition statement

2) I don’t see that they have deposited data on GenBank or plan to do so..

3) I saw no reference to an ethical approval statement

4) I saw no reference to field study permits

Reviewer 3 ·

Basic reporting

The English is good, but not great. It is obvious that the authors took great care to write professional, easy-to-read English, however, it is also clear that English is not their first language. As someone who has to write professionally at times in another language that is not my native English, I sympathize. It is hard.

I tried to make as many comments, suggestions, etc. as possible in my in-one comments. However, I would recommend that the manuscript get one more pass from a professional style editor.

Otherwise, the paper's reporting is fine. References and background are sufficient. Figures and tables are attractive and legible. The authors made attempts to make raw data available. However, unedited sequence data should be submitted to Genbank (not to FigShare); alignments (e.g. Nexus files) are best placed on FigShare. This is mandatory for the paper to be published.

Experimental design

The experimental design is solid. It is not cutting edge (which would be a more comprehensive genome-wide assessment of diversity), but it is solid and would have been state of the art a few years ago. The technical details are sound and the methods would allow replication.

However, Table S1 is insufficient and outside standards in our field. Actual museum voucher numbers (or tissue numbers) should be provided, rather than a summary of samples. And again, these should be provided with the Genbank accession by individual sample. Likewise, it would be nice to list the museum voucher numbers in a Table S2 for all specimens used in the morphological study. I realize it is a lot, but it helps for reproducibility. And it demonstrates value for scientific collections. This is perhaps more optional. Fixing Table S1 is not optional.

Validity of the findings

The findings are valid. However, I wish that the Discussion was more exciting. Simply talking about the role of the Pleistocene on bird diversification seems a little vanilla in 2018.

In general, I find that the Discussion lacks "hard thinking". It is as if the authors would prefer that the reader do all the cognitive work. Many papers take that approach, but better papers do not.

For example: what are the implications for the authors' findings for taxonomic limits in this group? The authors briefly bring this up, but do provide any guidance. Should the American Ornithological Society split this species? If so, where? If not, why?

Likewise, I wish that the authors would take more time to consider comparative biogeography. Why does this lineage has this pattern of isolation, but other co-distributed bird groups have other patterns of diversification across this space. Is there a role for ecology in explaining this pattern? Discussion of the importance of the Isthmus of Tehuantepec for highland birds (in the Discussion) is lacking.

Finally, what do you think that the L rhami group was doing between 10.5 MYA, which the lineage split from ancestors, and 600 kYA, when the current lineages diversified? It is speculation, but it would be valuable as a thought exercise for considering the conservation value of these lineages? If the pattern is that isolated groups are periodically formed, but also periodically go extinct, what should the conservation goal be? And if dispersal limitation drives diversification in this group, how did the ancestral taxon get to all these isolated ranges?

Additional comments

The abbreviations SMO, SMSn, SMS, and EITn are confusing. They take a lot of time to figure out. I would recommend using other abbreviations, even if it takes additional letters.

I have made many comments directly on the PDF. They are mostly about improving the readability of the manuscript.

Also, a minor note. Since 1983, the AOS no longer considers any subspecies. So line 96 is incorrect in that AOS doesn't consider geographic variation. The current AOS position on geographic races is that they refer you to Peters (1945). I would modify the spatement in line 96.

Annotated reviews are not available for download in order to protect the identity of reviewers who chose to remain anonymous.

---

## Round 0.2 · Minor Revisions

I have received comments from two reviewers, and they are happy with the new version. One of them made a few comments that I think can improve your paper. Please review them and send me the updated version as soon as possible.

·

Basic reporting

I have re-evaluated the revised version of the paper “Genetic and morphometric divergence in the Garnet-Throated Hummingbird Lamprolaima rhami (Aves: Trochilidae)” and I consider this new version improved substantially from the previous one.
The authors have made a number of positive changes to the paper to deal with my comments. These included rewriting several key sections and reworking Figures to add clarity. The flow of the paper is much clearer now and I believe it will be easily understood by readers.

Experimental design

The authors have made all the necessary changes to the paper to deal with my comments.

Validity of the findings

The authors have made all the necessary changes to the paper to deal with my comments.

Additional comments

I think the information provided in this manuscript will serve to provide further evidence to the role of the diversification of Mesoamerica biota. I am satisfied with the changes made by the authors and I consider the article more robust and clear.

Reviewer 3 ·

Basic reporting

I initially reviewed this manuscript as reviewer 3. In that review I found the findings sound, but identified areas where the interpretation could be improved, and also found the English quite difficult to follow. I consider this the most improved manuscript that I have ever re-reviewed. The authors have taken extraordinary care to improve their manuscript and should be commended. I believe that this manuscript is acceptable for publication. Below I highlight a few areas where I think that the manuscript could be improved, but I would consider these suggestions rather than mandates.

Experimental design

No concerns.

Validity of the findings

No concerns.

Additional comments

All these are suggestions, and quite minor. I do not need to re-review this manuscript. All these issues can be addressed editorially:

1. The abstract is a bit long. The authors should focus on what is most important from their work.

2. Perhaps the biggest issue for the paper: the authors seem to to conflate a checklist's decision to not consider subspecies with those authors considering that there is no geographical variation in the species. The AOU (now AOS) expanded from North America to North and Middle America (i.e West Indies and Central America) in 1983. When they did that, they decided to no longer consider subspecies across the board. So their decision isn't really a statement that Lamprolaima rhami doesn't not have subspecies. The situation with Gill and Donsker is less clear. They mention a few subspecies, but are inconsistent. So, in general, I would revisit the sentences on lines 99 and 463-464.

3. Line 399: perhaps "is better considered a hard rather than soft barrier...."

4. Lines 478-483 could be written with more impact. The authors seem to pull their punches afraid of (some) contemporary ornithologists dislike for subspecies. This is an opportunity to be more bold. The authors have done some great analyses to demonstrate a lack of meaningful gene flow and evolutionary distinctiveness of populations within the species. These three lineages may be species (the authors haven't really done a test one way or another) and even if they are not, they are evolutionarily-distinctive and deserve to be recognized as such. Since speciation begins with allopatric divergence, these three lineages have the potential to be future species; the authors should say that.

Kudos on a great revision.

---

## Round 0.3 · accepted · Accept

Congratulations! Please work with our production team to get your paper published.

#